# What happens to the inhibitory control functions of the right inferior frontal cortex when this area is dominant for language?

Esteban Villar-Rodríguez*, Cristina Cano-Melle, Lidón Marin-Marin, Maria Antònia Parcet, César Avila*

Neuropsychology and Functional Neuroimaging; Jaume I University, Castellón de la Plana, Spain

**Abstract** A low number of individuals show an atypical brain control of language functions that differs from the typical lateralization in the left cerebral hemisphere. In these cases, the neural distribution of other cognitive functions is not fully understood. Although there is a bias towards a mirrored brain organization consistent with the Causal hypothesis, some individuals are found to be exceptions to this rule. However, no study has focused on what happens to the homologous language areas in the right frontal inferior cortex. Using an fMRI-adapted stop-signal task in a healthy non right-handed sample (50 typically lateralized and 36 atypically lateralized for language production), our results show that atypical lateralization is associated with a mirrored brain organization of the inhibitory control network in the left hemisphere: inferior frontal cortex, presupplementary motor area, and subthalamic nucleus. However, the individual analyses revealed a large number of cases with a noteworthy overlap in the inferior frontal gyrus, which shared both inhibitory and language functions. Further analyses showed that atypical lateralization was associated with stronger functional interhemispheric connectivity and larger corpus callosum. Importantly, we did not find task performance differences as a function of lateralization, but there was an association between atypical dominance in the inferior frontal cortex and higher scores on schizotypy and autistic spectrum traits, as well as worse performance on a reading accuracy test. Together, these results partially support the Causal hypothesis of hemispheric specialization and provide further evidence of the link between atypical hemispheric lateralization and increased interhemispheric transfer through the corpus callosum.

**\*For correspondence:**
esvillar@uji.es (EV-R);
avila@psb.uji.es (CA)

**Competing interest:** The authors declare that no competing interests exist.

## eLife assessment

This study has **important** implications for theoretical proposals concerning how language lateralization affects the lateralization of other cognitive functions. The methods are **solid**, with an appropriate selection of cognitive control tasks that share homotopic regions of the brain with language, comparing participants with typical and atypical organization of language. The participants included in the study were mainly bilinguals, a population previously reported to have a more bilateral organization of cognitive control regions than monolinguals, limiting the generalizability of the results to the general population. Despite this limitation, the results will be of interest to researchers working of brain plasticity and development, in addition to those interested in language and cognitive control.

## Introduction

One of the oldest findings in human neuroscience is that the brain control of language is lateralized to the left hemisphere (***Broca, 1861***; ***Wernicke, 1874***; ***Price, 2012***). However, not all individuals present a typical organization of this function. Some left-handers (22–24%) show an atypical lateralization of language – right or ambilateral (***Pujol et al., 1999***; ***Szaflarski et al., 2002***; ***Mazoyer et al., 2014***). Also, this atypical organization is observed in some neurodevelopmental disorders such as schizophrenia, dyslexia, or autistic spectrum (***Bishop, 2013***; ***Sommer et al., 2001***; ***Eyler et al., 2012***).

How these atypical individuals organize the rest of their lateralized functions has been an extensively researched question. Historically, two different hypotheses have been proposed to explain the hemispheric specialization of the brain. The Causal hypothesis (***Kosslyn, 1987***; ***Hellige, 1990***) argues that rapid activities requiring the implementation of sequences of cognitive processes are better performed from a single, unilateral control. As a result, these activities are innately programmed to be lateralized, and, according to this model, some functions will be performed better if they are controlled by different hemispheres. This evolutionary pressure engenders a complementary relationship between these lateralized functions. For example, language production predominantly lateralizes to the left hemisphere, while visuospatial attention typically favors the right hemisphere. As per the Causal hypothesis, if language were to atypically lateralize to the right hemisphere, visuospatial attention would, in turn, shift to the left hemisphere. The Statistical hypothesis (***Bryden et al., 1983***), on the other hand**,** postulates that each cognitive function lateralizes independently from the others. In this view, the likelihood of encountering atypical hemispheric dominance is essentially a statistical phenomenon, with no inherent interrelationship between various lateralized functions. The evidence supporting each theory in left-handers is mixed. Some studies have obtained data favoring the Statistical hypothesis when analyzing language, visuospatial attention, and face processing (***Whitehouse and Bishop, 2009***; ***Badzakova-Trajkov et al., 2010***; ***Rosch et al., 2012***). Most investigations, however, have demonstrated a mirrored organization of praxis (***Vingerhoets et al., 2013***; ***Gerrits et al., 2020***), visuospatial attention (***Badzakova-Trajkov et al., 2010***; ***Gerrits et al., 2020***; ***Cai et al., 2013***), face recognition (***Badzakova-Trajkov et al., 2010***; ***Gerrits et al., 2020***; ***Gerrits et al., 2019***), and emotional prosody (***Gerrits et al., 2020***) in most individuals with atypical lateralization of language. Nonetheless, it should be noted that this reversed pattern is not observed in all atypical participants, and all the studies found specific cases in favor of both theories (***Gerrits et al., 2020***).

All these studies have employed a production task to investigate language lateralization in the inferior frontal cortex (IFC), classifying left-handed participants into typical and atypical groups. The rest of the lateralized cognitive functions have been examined using several tasks, but they all mainly depended on posterior parts of the brain. This may have limited the results, given that they were not dealing with directly homotopic functions, which would have been the most sensitive scenario to the proposed advantages of hemispheric specialization (***Rogers, 2000***; ***Vallortigara and Rogers, 2020***). To directly address the language production function, which is typically dependent on the left IFC, we would have to use a cognitive task that is typically dependent on the homotopic right IFC. According to previous literature, this function would be inhibitory control. Extensive results support the notion that stopping an ongoing response involves a rightward network that includes the IFC, presupplementary motor area (preSMA), and subthalamic nucleus (STN) (***Aron and Poldrack, 2006***; ***Xue et al., 2008***; ***Aron et al., 2014***; ***Hannah and Aron, 2021***). Thus, inhibitory control and language production together are the ideal candidates to investigate the Causal vs. Statistical disjunctive. But surprisingly, no investigation to date has analyzed this.

What causes language function to typically or atypically lateralize also remains unclear. Left-handedness appears to be, at best, a predisposing factor (***Mazoyer et al., 2014***). In clinical populations, atypical lateralization of language has been found to be a defining characteristic among patients of dyslexia (***Bishop, 2013***), schizophrenia (***Sommer et al., 2001***), and autism spectrum disorders (***Eyler et al., 2012***). However, aside from exhibiting poorer language performance (***Mellet et al., 2014***; ***Powell et al., 2012***), no study to date has identified subclinical markers of these neurodevelopmental conditions among healthy individuals with atypical lateralization. Regarding the neural basis of language lateralization, the corpus callosum hypothesis (***Gazzaniga, 2000***)— which suggests that typical hemispheric specialization relies on the proper development of this structure — has garnered attention in recent years due to several noteworthy findings: (1) regional functional lateralization is directly linked to the size reduction of its interconnecting corpus callosum segments (***Karolis et al.,***

*2019*); and (2) atypical lateralization of language has been associated with a larger corpus callosum and increased interhemispheric transfer at rest in language-relevant regions (*Labache et al., 2020*).

The primary aim of this study was to investigate the relationship between the lateralization of language production and inhibitory control. To achieve this, we divided non-right-handed participants into typically lateralized and atypically lateralized for language using the verb generation task. These two groups then completed the stop-signal task. In alignment with the predictions of the Causal hypothesis, we expected that: (1) the atypical group would exhibit a mirrored brain organization — particularly in the IFC, preSMA, and STN — during the stop-signal task; and (2) no individual would demonstrate lateralization of both language production and inhibitory control in the IFC of the same hemisphere. The secondary aim of this study was to explore certain potentially causal correlates of the lateralization of both functions. To test the presence of correlates supporting the corpus callosum hypothesis of lateralization, we analyzed the callosal volume and the interhemispheric functional connectivity at rest of the IFC, preSMA, and STN. To test the relation between atypical lateralization and several neurodevelopmental disorders (dyslexia, schizophrenia, and autism), we analyzed subclinical traits of these conditions among our healthy participants.

## Results
### Inhibitory control shifts its lateralization according to language production

Inhibitory control components during the stop-signal task were defined using four Regions Of Interest (ROIs) and subsequently analyzed through a MANOVA. This analysis included Hemisphere and Region

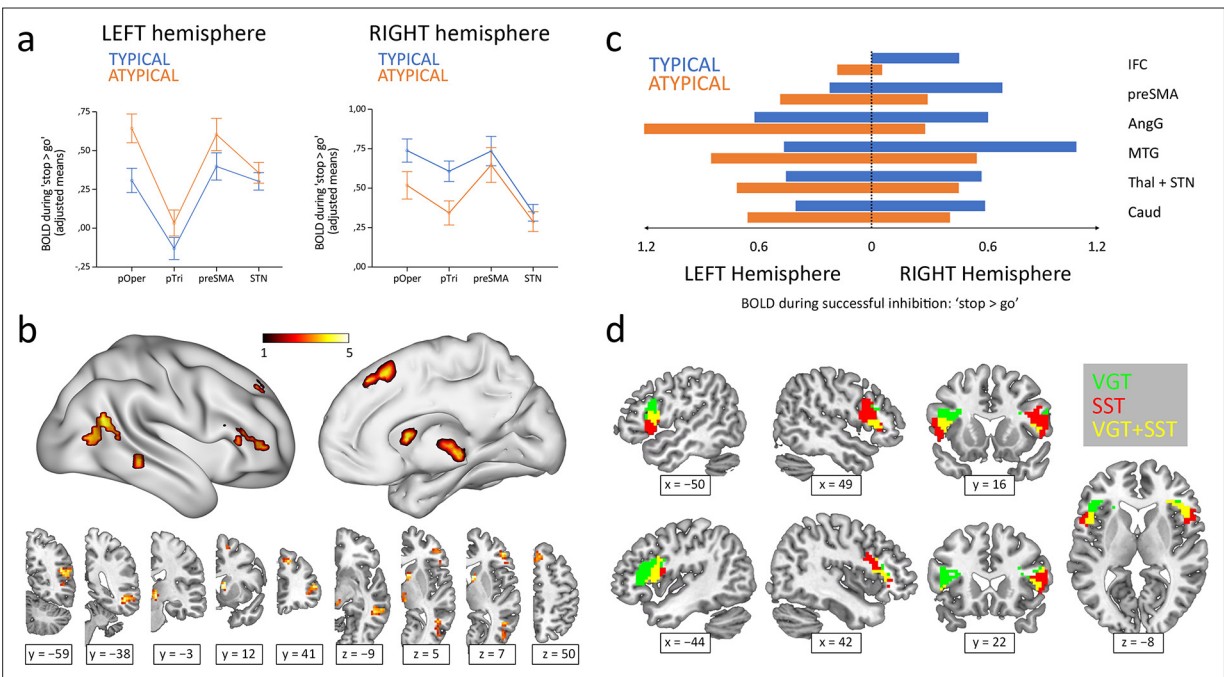

**Figure 1.** Hemispheric lateralization of inhibitory control according to language lateralization. (**a**) ROI analysis of the main components of the inhibitory control network. Graphs depict adjusted mean BOLD signal during successful inhibitions on the stop-signal task ('stop >go' contrast) for both hemispheres and both groups (n=50 typical and 36 atypical). All four structures showed significant Hemisphere × Group interactions in a repeated-measures MANOVA (p<0.05). Error bars represent one standard error. pOper = pars opercularis, pTri = pars triangularis, preSMA = presupplementary motor area, and STN = subthalamic nucleus. (**b**) Voxel-wise whole-brain analysis of functional asymmetry. Significance maps (voxel-wise p<0.001; FWE cluster-corrected at p<0.05; color bar represents t value) are displayed in three-dimensional reconstructions plus coronal and transversal slices using MNI space. (**c**) Mean BOLD values of the significant regions found in the voxel-wise whole-brain analysis. Graphic depicts BOLD values for every region, hemisphere, and group. IFC = inferior frontal cortex, preSMA = presupplementary motor area, AngG = angular gyrus, MTG = middle temporal gyrus, Thal = thalamus, STN = subthalamic nucleus, Caud = caudate. (**d**) Functional overlap between language production and inhibitory control in the IFC of ambilateral participants. Overlapping maps for inhibition (voxel-wise p<0.001; FWE cluster-corrected at p<0.05) and language (voxel-wise p<0.05; uncorrected) are displayed in coronal, sagittal and transversal slices using MNI space. VGT = verb generation task, SST = stop-signal task.

as within-subject factors and Group (typical/atypical) as a between-subject factor. Examination of the simple effects across each level of Region revealed that all the tested structures displayed a statistically significant Hemisphere × Group interaction (pars opercularis $F_{1,84}=25.59$, p<0.001; pars triangularis $F_{1,84}=19.95$, p<0.001; preSMA $F_{1,84}=7.63$, p=0.028; STN $F_{1,84}=14.24$, p<0.001; p values Bonferroni-adjusted; see *Figure 1a*). In other words, depending on the lateralization group, all four regions appeared to exhibit a leftward (atypical group) or rightward (typical group) lateralization of their BOLD signal during the 'correct stop >correct go' condition of the stop-signal task.

A voxel-wise whole-brain analysis of the functional asymmetry maps during the stop-signal task was performed to complement the ROI results. Two-sample *t*-test comparisons of the typical and atypical groups during the 'correct stop >correct go' condition showed a differential asymmetry pattern in all the ROI-corresponding areas, while also revealing differences in other cortical and subcortical regions, including the angular gyrus, middle temporal gyrus, and caudate (*Figure 1b*). Specifically, the typical

**Figure 2.** Correlation between the LIs of the verb generation and stop-signal tasks. *r* = −0.583, two-tailed p<0.001, $R^2$=0.339. Negative values indicate rightward lateralization, whereas positive values indicate leftward lateralization. Segregated and ambilateral phenotypes are also depicted according to the background color. The green area corresponds to segregated individuals (both functions strongly lateralized), and the red area corresponds to integrated individuals (at least one function ambilaterally controlled). Numbers inside each quadrant denote the number of individuals (n) contained in it. Each individual datapoint is symbolized according to its functional organization: ○=typical segregation; ●=reversed segregation; △=ambilateral inhibition; ▽=ambilateral language; ◇=ambilateral language and inhibition.

**Table 1.** Spearman's partial correlations between task LIs and neuroanatomical plus behavioral variables.

| | Verb generation task LI | | Stop-signal task LI | |
|---|---|---|---|---|
| | Spearman's $\rho$ | p | Spearman's $\rho$ | p |
| Callosal genu volume | −0.249 | 0.021* | 0.251 | 0.02* |
| Callosal body volume | −0.258 | 0.017* | 0.132 | 0.224 |
| Callosal splenium volume | −0.249 | 0.021* | 0.118 | 0.279 |
| Pars Opercularis VMHC | −0.062 | 0.584 | 0.148 | 0.192 |
| Pars Triangularis VMHC | −0.119 | 0.295 | 0.258 | 0.022* |
| preSMA VMHC | 0.052 | 0.647 | 0.037 | 0.748 |
| STN VMHC | 0.033 | 0.772 | 0.217 | 0.054 |
| 'Go' reaction time | 0.125 | 0.251 | −0.041 | 0.71 |
| 'Go' accuracy | −0.076 | 0.489 | 0.039 | 0.724 |
| SSRT | 0.173 | 0.111 | −0.038 | 0.726 |
| Reading length accuracy | −0.237 | 0.028* | 0.214 | 0.047* |
| Reading familiarity accuracy | −0.232 | 0.032* | 0.301 | 0.005** |
| Reading length time | 0.054 | 0.622 | −0.184 | 0.068 |
| Reading familiarity time | 0.091 | 0.403 | −0.198 | 0.091 |
| SPQ | −0.194 | 0.078 | 0.247 | 0.023* |
| AQ | −0.247 | 0.023* | 0.075 | 0.498 |

General intelligence and age were included as covariates of no interest. Callosal volume correlations were additionally corrected for total intracranial volume.

group presented a rightward activity in these regions, whereas the atypical group exhibited a leftward shift (*Figure 1c*).

Finally, the Pearson's correlation between the frontal LIs (pars opercularis + pars triangularis) from the verb generation task and the stop-signal task revealed a strong inverse correlation ($r_{84}$ = −0.58; two-tailed p<0.001; $R^2$=0.34) (*Figure 2*). Out of 50 typically lateralized for language, 33 (66%) presented a typical right lateralization for inhibitory control, and 17 (34%) showed an atypical (left – ambilateral) lateralization. Among the 36 individuals from the atypical group, only 8 (22.2%) presented a typical lateralization during the stop-signal task, whereas 28 (77.8%) showed an atypical lateralization. Importantly, no participant presented a leftward + leftward or rightward + rightward segregation pattern for language production and response inhibition. Strong segregation of both functions into different hemispheres was observed in 38 participants (44.2%). The remaining 48 individuals (55.8%) presented an ambilateral representation of one or both functions, with an important level of spatial overlap in their activations (see *Figure 1d*). In this ambilateral group, language production presented an overlap in 32.3% of its voxels in the left hemisphere, and in 58.6% of its voxels in the right hemisphere. Inhibitory control showed an overlap in 57.2% of its voxels in the left hemisphere, and in 22.8% of its voxels of the right hemisphere.

## Laterality indexes correlate to interhemispheric functional connectivity, callosal volume and preclinical markers

Correlational analyses revealed certain links between the functional lateralization of both tasks and behavioral, neuroanatomical and connectivity measures (*Table 1*).

First, we studied how the inter-hemispheric functional connectivity and callosal size behaved in relation to LI. Volume in the genu, body and splenium of the corpus callosum was inversely related with language LI (respectively: $\rho_{83}$ = −0.25, p=0.02; $\rho_{83}$ = −0.26, p=0.02; and $\rho_{83}$ = −0.25, p=0.02). However, only the callosal genu was found related with inhibition LI ($\rho_{83}$=0.25, p=0.02). That is, the volume of the callosal genu increased as functional organization of the IFC became more atypical,

extending this effect to the whole corpus callosum when considering exclusively language lateralization. On the same line, ROI interhemispheric functional connectivity analyses at rest revealed that VMHC of the pars triangularis also increased as a function of inhibition LI ($\rho_{77}$=0.26, p=0.02), but a similar relation failed to be found for language LI ($\rho_{77}$ = −0.12, p=0.3). No statistically significant linear relationships were found when exploring the VMHC of the pars opercularis, the preSMA, or the STN. It should be mentioned, however, that the association between VMHC of the STN and inhibition LI was close to significance ($\rho_{77}$=0.22, p=0.054).

Behaviorally, performance (RT, SSRT, and accuracy) during the scanner stop-signal task did not correlate with language and inhibition LIs. Regarding the reading test, accuracy for word length and word familiarity were found correlated with both LIs (length and language: $\rho_{84}$ = −0.24, p=0.03; length and inhibition: $\rho_{84}$=0.21, p=0.04; familiarity and language: $\rho_{84}$ = −0.23, p=0.03; familiarity and inhibition: $\rho_{84}$=0.3, p=0.01). That is, an atypical organization was related with a higher rate of errors when reading long words and unfamiliar words. No significant correlations were found when exploring reading speed. SPQ score was found significantly related with inhibition LI ($\rho_{82}$=0.25, p=0.02), but it did not reach statistical significance when paired with language LI ($\rho_{84}$ = −0.19, p=0.08). AQ, on the other hand, presented an association with language LI ($\rho_{84}$ = −0.25, p=0.02) but not with inhibition LI ($\rho_{84}$=0.0.8, p=0.5).

## Discussion

The present study employed the stop-signal task to investigate brain areas involved in the inhibitory control function in two different groups of individuals: typically and atypically lateralized for language production. As expected, the atypical participants showed, as a group, a mirrored brain organization of the inhibitory control function compared to the typical group. This leftward organization affected the entire inhibitory control network, including the IFC, preSMA, and STN. However, some participants manifested a clear overlap in the control of language and inhibitory functions. Our results also demonstrate that atypical organization of language production is associated with an increased white matter volume of the corpus callosum, and that atypical lateralization of inhibitory control is related with a higher interhemispheric functional coupling of the IFC. Behaviorally, atypical lateralization was not associated with performance on the task, but an association existed with worse reading accuracy, and more schizotypy and autistic traits.

Our first relevant result is that the group of individuals with atypical lateralization of language presented a mirrored brain organization during the stop-signal task compared to the typically lateralized group. Remarkably, the hemispheric asymmetry differences obtained affected the entire inhibitory control brain network (*Hannah and Aron, 2021*). The inferior frontal cortex, or IFC, associated with the initiation of the stop command and the main cortical area of this system, showed a clear lateralization effect, with the typical group almost exclusively using the right part and the atypical group mostly activating the left part. The presupplementary motor area, or pre-SMA, related to the implementation of the stop command, was more bilaterally involved in both groups, although atypicals activated the left part more and typicals activated the right part more. Importantly, the hemispheric reversal of organization also affected the subcortical structures participating in this network, namely the STN and thalamus. These structures displayed a lateralization pattern similar to the preSMA, with a more pronounced bilateral engagement but higher involvement of the right part in typicals and of the left part in atypicals. Thus, this is the first demonstration that an entire cerebral network controlling a cognitive function shows a completely flipped organization in an atypical population for language, which supports the Causal hypothesis of lateralization (*Kosslyn, 1987*; *Hellige, 1990*). Moreover, this result reveals that hemispheric lateralization goes beyond cerebral cortices, and even interhemispheric connectivity, through the corpus callosum, given that it also involves frontal-subcortical circuits. Additionally, the specificity of this result to the areas of the proposed network is another element that supports the relevance of this system in inhibitory control.

It should be noted that previous studies have demonstrated a significantly higher involvement of the left IFC during cognitive control tasks in bilingual individuals when compared to monolinguals (*Garbin et al., 2010*; *Rodríguez-Pujadas et al., 2013*). However, this could not be directly considered in the current study due to the limited number of monolinguals participants (n=8), which would not allow for statistically robust tests. So, we cannot rule out the possibility that bilingualism could be acting as an enabling mechanism among atypically lateralized participants for the shift of

the inhibitory control network to the left hemisphere. Therefore, the findings of this study should be extrapolated to the monolingual population with caution.

Remarkably, this pattern of reversed specialization for inhibitory control was observed in the absence of a relation between functional lateralization and performance differences during the 'go' or 'stop' conditions. Even though the 'go' condition involves a lexical decision requiring language processing, and the 'stop' condition consists of inhibiting a linguistic process, the functional organization of the IFC during the tasks did not affect the response speed (RT and SSRT) or the 'go' accuracy in either case. Thus, brain organization for language and inhibition, as initially defined in our study, and cognitive efficiency are not directly related, based on our data. This finding is consistent with prior studies highlighting that: (1) a densely segregated functional organization, rather than a reversed organization, is linked to general (*Gerrits et al., 2020*) and specific (*Mellet et al., 2014*) cognitive deficits; and (2) specific cognitive deficits appear to be confined to the domains of verbal and spatial skills (*Mellet et al., 2014*).

Importantly, despite observing a complimentary relationship between language production and inhibitory control at the group level, not all the individuals presented a strong and segregated distribution of these two functions. The vast majority of hemispherically segregated individuals showed the typical segregation of language in the left hemisphere and inhibitory control in the right hemisphere, whereas only a low percentage displayed the opposite segregated organization. The rest, in agreement with similar reports on other functions (*Gerrits et al., 2019*), lacked a clear hemispheric lateralization of language production, inhibitory control, or both. In all cases, this implied some level of overlapping or sharing of the same area of the inferior frontal cortex. Both the low ratio of atypical segregation and the high ratio of bilateral inhibitory control could be explained by the fact that typical right hemispheric functions (such as inhibition) seem to be more bilaterally represented than typical left functions (such as language; *Gotts et al., 2013*). Some conclusions can be extracted from this: (1) Although infrequent, a strong atypical segregation of the two functions is possible; (2) Our data have not revealed any case with a segregation of both frontal cognitive functions in the same hemisphere; and (3) A large percentage of participants with a strong left lateralization of language presented a bilateral control of inhibition, showing that a typical dominance for language does not imply a strongly lateralized organization of other cognitive functions. The existence of exceptions to the group pattern of mirrored brain organization requires us to use caution when interpreting these data as supporting the causal hypothesis. Although the negative correlation between the lateralization indexes during the verb generation task and the stop-signal task is strong at the group level, some individual data support the statistical hypothesis (*Bryden et al., 1983*) instead. Given that we found no cases of both cognitive functions completely sharing a hemisphere (see *Figure 2*), we propose that causal-supporting vs. statistical-supporting results reflects two different pathways to cognitive control. On the one hand, the segregated pathway (in line with the Causal hypothesis) develops due to the cerebral bias towards lateralizing certain cognitive functions in a single hemisphere. On the other hand, the integrated pathway (in line with the Statistical hypothesis), appears when ontogenetic development towards strong lateralized control fails in some way.

These findings could have significant implications when studying neonatal lesions that facilitate the development of left-handedness (*Dinomais et al., 2017*), or when assessing the consequences of brain lesions in the left-handed population (*Newport et al., 2022*). For example, a left-sided lesion in an individual with rightward language lateralization may result in inhibitory control and social deficits (*Mosch et al., 2005*). It also raises questions about neuroplasticity, particularly how functional reorganization following a brain lesion (such as a left-to-right reorganization of language) would impact this hemispheric relationship between language and inhibitory control, and the cognitive consequences for inhibitory control after such a reorganization (*Bates et al., 2001*). These results could also offer valuable insights for cognitive rehabilitation procedures in patients with psychiatric disorders that seem to be linked to a higher prevalence of left-handedness and alterations in language lateralization (*Webb et al., 2013*).

The corpus callosum is the main cerebral structure in interhemispheric connectivity, with the genu area being responsible for the connectivity between the two inferior frontal gyri (*Hofer and Frahm, 2006*). Phylogenetic data have demonstrated a negative association between the corpus callosum volume and hemispheric specialization, suggesting that the ontogenetic development of the brain is supported by a decrease in interhemispheric connectivity to potentiate and establish intrahemispheric

connectivity and hemispheric specialization (*Tzourio-Mazoyer, 2016*). There is evidence of this mechanism in language development (*Tzourio-Mazoyer and Seghier, 2016*) and in agenesis of the corpus callosum, which has been associated with lower lateralization of language and worse language performance (*Hinkley et al., 2016*; *Bartha-Doering et al., 2021*). In line with this proposal, we have replicated the relation between callosal enlargement and atypical functional organization found in a previous study (*Labache et al., 2020*). Moreover, in our data, this structural correlation was accompanied by an association between atypical lateralization of inhibitory control and strength of interhemispheric functional connectivity in the pars triangularis of the IFC.

As hypothesized, when considering the functional lateralization during both tasks, atypical organizations were found linearly related with higher scores on a schizotypy test (only for inhibitory control), higher scores on an autistic spectrum test (only for language production), and a higher rate of reading errors (both language production and inhibitory control). The first result is consistent with previous structural, functional, and behavioral data that have shown reduced language lateralization in schizophrenia patients (*Sommer et al., 2001*; *Li et al., 2007*). In the case of schizotypy, the evidence for reduced lateralization of cognitive functions is mixed and may respond to differences in the methodology employed (*Park and Waldie, 2017*). However, consistently with the present results, this condition has been previously associated to impaired behavioral and neural processing during the stop-signal task (*Jia et al., 2021*), and to an abnormal frontal functional asymmetry (*Le et al., 2020*). The second result relates atypical language lateralization with the presence of autistic traits, a result consistent with data obtained in autism (*Jouravlev et al., 2020*). The third result shows that atypical organization was related with a higher rate of reading errors when considering word length and word familiarity. The length effect has been considered a pathognomonic symptom of reading disorders such as developmental dyslexia (*Provazza et al., 2019*) and pure alexia (*Roberts et al., 2013*). These disorders have also been characterized by a weak language lateralization (*Bishop, 2013*). Thus, presented results support the hypothesis that atypical hemispheric specialization is related with worse cognitive performance in the linguistic domain, and even preclinical traits of some neurodevelopmental disorders among healthy population.

In conclusion, our study demonstrates a strong connection between the lateralization of language production in the IFC and the hemispheric specialization of the inhibitory control network. This functional interrelationship is apparent through both regional and voxel-wise analyses and impacts all the primary cerebral hubs of inhibitory control. Furthermore, the atypical lateralization of these two cognitive functions shows a linear correlation with increased inter-hemispheric functional connectivity in the IFC, a larger corpus callosum, the presence of subclinical markers of schizotypy and autism, and decreased reading accuracy. Consequently, our research provides compelling evidence in support of the causal and corpus callosum hypotheses of lateralization among left-handers, and offers valuable insights into the pathogenic mechanisms underlying atypical lateralization in healthy population.

## Materials and methods
### Participants
Eighty-six participants were included in the present study. They were selected following a functional magnetic resonance imaging (fMRI) language lateralization assessment via a verb generation task. Hence, 50 were typically lateralized for language – left-dominant – (mean ± SD age=22.4 ± 3.6 years; 24 male, 26 female) and 36 were atypically lateralized (mean ± SD age=23.4 ± 4 years; 16 male, 20 female).

All the participants were non-right-handed, according to the Edinburgh Handedness Inventory (*Bryden, 1977*; *Oldfield, 1971*). There were no significant between-group differences in age ($t_{84}$=1.28; p=0.2), sex ($\chi^2$=0.11; p=0.74), or general intelligence by WAIS-IV matrix reasoning subtest (*Wechsler, 2012*; $t_{84}$=1.8; p=0.08). The participants had no history of any neurological or psychiatric disorders or head injury with loss of consciousness. Written informed consent was obtained from all participants following a protocol approved by Universitat Jaume I. All methods were carried out in accordance with approved guidelines and regulations.

Regarding their bilingual status, we found three categories: Spanish monolinguals (n=6 typical and 2 atypical), Spanish-Catalan bilinguals (n=27 typical and 23 atypical); and Spanish-Catalan passive bilinguals (n=17 typical and 11 atypical). Passive bilingualism refers to the fact that some residents of

this region (Valencian Community) understand Spanish and Catalan, but their frequency of use of one language is extremely low when compared to the other. No significant between-group differences were found in bilingual status when comparing the typical and atypical groups ($\chi^2$=1.2; p=0.55).

## Experimental design

Participants were recruited via multiple advertisements across Castellón and Valencia universities (bulletin boards, mass emailing, etc.) asking for the collaboration of left-handers in an fMRI brain study. Persons older than 36 years or younger than 16 years were discarded. Valid participants were cited for a first fMRI session, in which they completed the verb generation task and the resting-state acquisition. During this session, we used the BrainWave software (GE HealthCare Technologies Inc) to visualize real-time data and roughly categorize participants as potentially typical or potentially atypical. Forty-three participants were found potentially atypical for language lateralization, and were invited to a second fMRI session, along with 43 potentially typical participants. During this second session, participants completed the stop-signal task (scanner) and a reading skill test (out of scanner). Also, data regarding schizotypy personality and autistic spectrum was gathered via self-questionnaires. Note that classification of participants as potentially typical or potentially atypical was used for screening purposes, and it did not completely match the final assessment.

### Verb generation task

Expressive language function was measured via fMRI during a computerized verb generation task (*Sanjuán et al., 2010*; *Villar-Rodríguez et al., 2020*) that consists of a two-block design paradigm with intercalating activation and control blocks. In the activation blocks, participants are asked to overtly say the first verb that comes to mind when visually presented with a concrete noun. In the control blocks, they have to read visually-presented letter pairs aloud. This task lasted for 6 min, with a block duration of 30 s (6 activation blocks and 6 control blocks), a stimulus duration of 1500 ms, and a blank inter-stimulus interval of 1500 ms. Before entering the scanner, participants practiced with a different version of the task for 1 min. Stimuli were presented using MRI-compatible goggles (VisuaStim Digital, Resonance Technology Inc), and responses were recorded with a noise-cancelling microphone (FOMRI III+, Optoacoustics Ltd.) to verify that each participant was engaged correctly in the task.

### Stop-signal task

Response inhibition was measured via fMRI during a computerized stop-signal task adapted from another study (*Xue et al., 2008*). The task consists of an event-related design paradigm with 'Go' and 'Stop' trials. In the 'Go' trials, participants are asked to manually answer by pressing a button if the visually presented string is a word (has a meaning) or a pseudoword (mimics typical word structure but has no meaning). Words require an index button press, whereas pseudowords require a thumb button press. However, on the 'Stop' trials, the string is followed by a 'beep' noise that signals to the participants to inhibit their response and refrain from pressing any button. The instructions emphasized the importance of going correctly and stopping correctly. However, participants were asked to respond as quickly and accurately as possible and avoid withholding their responses in anticipation of a possible 'beep'.

This task took 13 min and 28 s, and it was divided into two runs that lasted 6 min and 14 s each, separated by a 1 min rest. Each run consisted of 135 trials, of which 32 (23.7%) were 'Stop' trials. All the trials randomly presented words or pseudowords. The trial structure consisted of a fixation cross-hair (500ms), a word or pseudoword (1000ms, during which the 'beep' may or may not be presented at some point), and a blank inter-stimulus interval (ranging from 500 to 4000ms, sampled from an exponential distribution truncated at 4000ms, with mean of 1000ms). The Stop Signal Delay or SSD (the amount of time after the onset of the word or pseudoword when the 'beep' is presented during 'Stop' trials) changed dynamically during the task after each 'Stop' trial, depending on whether inhibition was successful (+25ms) or unsuccessful (−25ms), with a minimum of 100ms and a maximum of 800ms. The lower the SSD, the easier it is to inhibit the response, and vice versa. Hence, a dynamic SSD normalizes the task's difficulty across all participants based on their performance, aiming at a 50% successful inhibition rate. The SSD used for the first 'Stop' trial was estimated for each participant based on their practice session before entering the scanner. This practice session lasted 6 min and used a different set of words and pseudowords. Stimuli were presented using MRI-compatible

goggles and headset (VisuaStim Digital, Resonance Technology Inc), and responses were recorded via an MRI-compatible response-grip (ResponseGrips, NordicNeuroLab). The 'beep' volume was kept at a comfortable level and was constant across participants.

## Resting-state

Functional connectivity was measured via fMRI during a resting-state session (*Biswal et al., 1995*). In this paradigm, participants were presented with a fixation crosshair and instructed to just lie in the scanner with their eyes open and try not to sleep or think about anything in particular. This session lasted for 7 min. Seven participants did not complete this session due to time constraints, and they were subsequently removed from the functional connectivity analyses. It is important to highlight that the exclusion of these seven participants across all analyses does not notably impact the overall results.

## Behavioral measures

Individual inhibition speed was estimated by calculating the Stop-Signal Reaction Time (SSRT; *Verbruggen et al., 2019*). SSRT was computed as the difference between the median reaction time (RT) on correct 'go' trials and the mean SSD on the stop-signal task. We chose to employ the median RT instead of the mean RT, in accordance with the approach used in the study from which we adapted our stop-signal task (*Xue et al., 2008*). This choice was made for two specific reasons: (1) when working with RT, using the median helps reduce the impact of outlier responses during the task; and (2) using the median RT allowed our sample's SSRT to align more closely with a normal distribution. It should be noted that technical problems involving the MRI-compatible response-grip invalidated the SSD and RT data of four participants, and so their SSRT was estimated using the data from the practice session instead. This practice session, which lasted for 6 min, took place outside of the scanner in a different room, but using an identical set of response-grip. So, for these four participants, the resulting SSD and RT after those 6 min of practice was used for their SSRT calculation.

Reading skill was evaluated using the word reading subtest of the PROLEC-SE-R (*Cuetos Vega et al., 2016*), a battery that assesses reading processes in Spanish. Participants had to read four lists of words consisting, respectively, of short familiar words, long familiar words, short unfamiliar words, and long unfamiliar words. Responses were recorded with a microphone, and their accuracy and speed were subsequently measured by computing length (performance during short words – performance during long words) and familiarity (performance during familiar words – performance during unfamiliar words) indicators.

Schizotypal traits were explored with the SPQ (*Raine and Raine, 1991*), a preclinical self-report questionnaire modeled after DSM-III criteria (*American Psychiatric Association, 1980*). This questionnaire evaluates schizotypy based on the three-dimensional model: disorganized traits, cognitive-perceptual traits, and interpersonal traits. For two participants, these data were not collected due to time constraints.

Autistic spectrum traits were explored with the AQ (*Baron-Cohen et al., 2001*), a preclinical self-report questionnaire modeled after DSM-IV criteria (*Association, 1994*). This questionnaire evaluates autistic spectrum based on the 'triad' of impaired communication, impaired social skills, and a restricted and repetitive way of acting (*Rutter, 1978*), plus two sub-scales for imagination and attention to detail. For two participants, these data were not collected due to time constraints.

## Image acquisition

Images were acquired on a 3T General Electric Signa Architect magnetic resonance imaging (MRI) scanner using a 32-channel head coil. All slices were acquired in the sagittal plane. A 3D structural MRI was acquired for each subject using a T1-weighted magnetization-prepared rapid gradient-echo sequence (TR/TE = 8.5/3.3ms; flip angle = 12; matrix = 512 × 512 × 384; voxel size = 0.47 × 0.47 × 0.5). For the fMRI, a gradient-echo T2*-weighted echo-planar imaging sequence was used in the acquisition of 150 functional volumes on the verb generation task (TR/TE = 2500/30ms; flip angle = 70; matrix = 64 × 64 × 30; voxel size = 3.75 × 3.75 × 4), and a different sequence was used in the acquisition of 374 functional volumes during the stop-signal task and 210 functional volumes in the resting-state (TR/TE = 2000/30ms; flip angle = 70; matrix = 64 × 64 × 27; voxel size = 3.75 × 3.75 × 4.5).

## Image processing

### Task-based fMRI processing

Task functional images were processed using the Statistical Parametric Mapping software package (SPM12; Wellcome Trust Centre for Neuroimaging, London, UK). Preprocessing followed the default pipeline and included: (a) alignment of each participant's fMRI data to the AC-PC plane by using the anatomical image; (b) head motion correction, where the functional images were realigned and resliced to fit the mean functional image; (c) co-registration of the anatomical image to the mean functional image; (d) re-segmentation of the transformed anatomical image using a symmetric tissue probability map; (e) spatial normalization of the functional images to the MNI (Montreal Neurological Institute, Montreal, Canada) space with 3 mm$^3$ resolution; and (f) spatial smoothing (FWHM = 4 mm).

The general linear model for the verb generation task was defined for each participant by contrasting *activation > control blocks*. The general linear model for the stop-signal task was defined for each participant by contrasting *correct 'stop' > correct 'go' trials*. For both tasks, the BOLD (Blood-Oxygen-Level-Dependent) signal was estimated by convolving the task's block/trial onsets with the canonical hemodynamic response function. Six motion realignment parameters extracted from head motion preprocessing were included as covariates of no interest, and a high-pass filter (128 s) was applied to the contrast images to eliminate low-frequency components.

After that, we utilized the VOI analysis function within SPM12 to extract the first eigenvariate of the BOLD time courses from the stop-signal contrast images, which highly resembles the mean time course across the voxels. This extraction was carried out region-wise for the main components of the inhibitory control network: pars opercularis, pars triangularis, preSMA, and STN. Pars opercularis and pars triangularis of the IFC were defined following the criteria of the Harvard-Oxford atlas (*Frazier et al., 2005*; *Desikan et al., 2006*; *Makris et al., 2006*; *Goldstein et al., 2007*). For the PreSMA region, we used the SMA region, as defined by the same atlas, but including only voxels anterior to MNI Y=0 (*Ruan et al., 2018*). STN was defined by a 10 mm box centered at MNI coordinates 10,–15, –5 (*Aron and Poldrack, 2006*; *Lucerna et al., 2002*).

Lastly, we computed whole-brain voxel-wise functional asymmetry maps from the stop-signal contrast images. To do so, stop-signal contrast images were flipped at midline, inverting the right and left hemispheres, and subsequently subtracted from the original unflipped contrast images (*Kurth et al., 2015*).

### Resting-state fMRI processing

Resting-state functional images were processed using the Data Processing Assistant for the Resting-State toolbox (DPARSFA; *Chao-Gan and Yu-Feng, 2010*), which is based on SPM and the Data Processing & Analysis of Brain Imaging toolbox (DPABI; *Yan et al., 2016*). Preprocessing steps included: (a) slice-timing correction for interleaved acquisitions; (b) head motion correction (no participant had a head motion of more than 2 mm maximum displacement in any direction or 2° of any angular motion throughout the scan); (c) co-registration of the anatomical image with the mean functional image; (d) new segmentation to DARTEL; (e) removal of nuisance variance through linear regression: six parameters from the head motion correction, white matter signal, cerebrospinal fluid signal, and global mean signal; (f) spatial normalization to the MNI (3 mm$^3$); (g) spatial smoothing (FWHM = 4 mm); (h) removal of the linear trend in the time series; (i) band-pass temporal filtering (0.01–0.1); (j) normalization to a symmetric template; (k) Voxel-Mirrored Homotopic Computation (VMHC) (*Zuo et al., 2010*), calculated as the Pearson correlation coefficient of every voxel with its hemispheric counterpart; and (l) normalization of all voxel-wise time courses to Fisher *z* values. Finally, mean VMHC values were extracted for the pars opercularis, pars triangularis, preSMA, and STN. ROI definition can be found in the previous section.

### Structural MRI processing

Structural images were processed via voxel-based morphometry (VBM) using the CAT12 toolbox (*Gaser and Kurth, 2016*), which is based on SPM. Preprocessing steps followed the recommended pipeline and included: (a) segmentation into grey matter, white matter, and cerebrospinal fluid; (b) registration to the ICBM standard template; (c) modulated normalization of grey matter and white matter segments to the MNI template; (d) spatial smoothing (FWHM = 6 mm); and (e) extraction of Regions of Interest (ROIs) values from native space. Three ROIs were delimited by the voxels mapped

as genu, body and splenium of the corpus callosum according to the Mori atlas (*Oishi et al., 2009*). Total intracranial volume was also extracted for use as covariate of no interest.

## Individual assessment of functional lateralization

Individual functional lateralization was assessed by calculating the Laterality Index (LI) on the unflipped contrast images (*Sanjuán et al., 2010*; *Villar-Rodríguez et al., 2020*). We used the bootstrap method implemented in the LI-toolbox (*Wilke and Lidzba, 2007*), based on SPM. LI is a proportion of the brain activation between the two hemispheres, thus giving us information about the direction and degree of hemispheric specialization during a particular function in a single individual. LI ranges from +1 (totally leftward function) to −1 (totally rightward function). For both the verb generation task and the stop-signal task, we explored the LI of the inferior frontal region roughly corresponding to the classic Broca's area: pars opercularis and pars triangularis of the inferior frontal gyrus, according to the Harvard-Oxford atlas (*Frazier et al., 2005*; *Desikan et al., 2006*; *Makris et al., 2006*; *Goldstein et al., 2007*). This region is critical for both language production (*Price, 2012*) and response inhibition (*Aron et al., 2014*), depending on the hemisphere. During the verb generation task, language production was classified as typically lateralized (LI higher than +0.4) or atypically lateralized (LI lower than +0.4). We used +0.4 as a cut-off point (contrary to the more traditional +0.2), based on previous findings that emphasized the importance of lateralization strength when grouping individuals (*Mazoyer et al., 2014*; *Labache et al., 2020*).

## Statistical analyses

### Task-based region-wise analysis

We performed different analyses to investigate the hypothesis that the atypical group would show a mirrored brain organization during the stop-signal task. First, we conducted a region-wise analysis to compare the functional asymmetry of the primary components within the inhibitory control network between the typically and atypically lateralized groups. To achieve this, we employed a repeated-measures MANOVA, with Hemisphere (left/right) and Region (pars opercularis/pars triangularis/preSMA/STN) as within-subject factors and Group (typical/atypical) as the between-subject factor. Subsequently, to assess potential hemispheric differences in any of these regions between both groups, we contrasted the simple effects of the Hemisphere × Group interaction across the Region factor. All resulting *p*-values were adjusted using a Bonferroni correction.

### Task-based voxel-wise analyses

Previous analysis was complemented with whole-brain functional asymmetry contrast maps (extracted from the stop-signal task) that were compared in typically and atypically lateralized groups. To do so, we performed a whole-brain two-sample *t*-test (voxel-wise p<0.001; FWE cluster-corrected at p<0.05) via SPM12.

We also explored the functional overlap between both tasks in the subgroup of participants that presented an ambilateral organization of at least one function (n=48). To do so, we overlayed two one-sample *t*-tests restricted to the IFC: one for the stop-signal task (voxel-wise p<0.001; FWE cluster-corrected at p<0.05), and one for the verb generation task (voxel-wise p<0.05; uncorrected). Functional overlap was then expressed as the proportion of overlapped voxels for every function and hemisphere.

### Correlational analyses

The relationship between the hemispheric specialization of the verb generation task and the stop-signal task at the IFC was also examined linearly. More precisely, we conducted a Pearson's correlation between the LI from the verb generation task and the LI from the stop-signal task used in the individual lateralization classification (p<0.05, two-tailed).

Using these LIs, we also tested if a linear relationship existed between the functional lateralization of an individual, and several behavioral tests and neuroanatomical measures. Partial Spearman's correlations were computed between the LI of both tasks and: stop-signal task performance (RT, accuracy, and SSRT), SPQ derived-scores, AQ derived-scores, reading test performance (length and familiarity effects on speed and accuracy), functional connectivity measures (VMHC), and white matter

volume of the callosal genu. Age and general intelligence were included as covariates in all analyses. Total intracranial volume was also included as covariate in the volumetric analysis.

## Acknowledgements

This work was supported by the Spanish State Research Agency (PID2019-108198GB-I00) and the Universitat Jaume I (UJI-B2021-11). EV-R was supported by a predoctoral graduate program grant from the Spanish Ministry of Education, Culture and Sports (FPU18/00687). We also thank all of our participants for their collaboration in this study, as well as the radiographers at the clinic ASCIRES Castellón for their technical support during data acquisition.

## Additional information

### Funding

| Funder | Grant reference number | Author |
|---|---|---|
| Agencia Estatal de Investigación | PID2019- 108198GB-I00 | César Avila |
| Universitat Jaume I | UJI-B2021-11 | César Avila |
| Ministerio de Educación y Formación Profesional | FPU18/00687 | Esteban Villar-Rodríguez |

The funders had no role in study design, data collection and interpretation, or the decision to submit the work for publication.

### Author contributions

Esteban Villar-Rodríguez, Data curation, Formal analysis, Investigation, Methodology, Writing – original draft, Writing – review and editing; Cristina Cano-Melle, Data curation, Investigation; Lidón Marin-Marin, Investigation; Maria Antònia Parcet, Conceptualization, Writing – review and editing; César Avila, Conceptualization, Supervision, Methodology, Writing – original draft, Project administration, Writing – review and editing

### Author ORCIDs

Esteban Villar-Rodríguez ⓘ https://orcid.org/0000-0001-9691-3776
César Avila ⓘ https://orcid.org/0000-0002-5840-605X

### Ethics

All methods were carried out in accordance with guidelines and regulations approved by the Research Ethics Committee of the Universitat Jaume I (reference number: CD/34/2019). Written informed consent was obtained from all participants, following a protocol approved by the Universitat Jaume I, and they received monetary compensation for their participation.

Reviewer #1 (Public Review): https://doi.org/10.7554/eLife.86797.3.sa1
Reviewer #2 (Public Review): https://doi.org/10.7554/eLife.86797.3.sa2
Author Response https://doi.org/10.7554/eLife.86797.3.sa3

## Additional files

### Supplementary files
• MDAR checklist

### Data availability
All data presented in this study are included in figshare.

The following dataset was generated:

| Author(s) | Year | Dataset title | Dataset URL | Database and Identifier |
|---|---|---|---|---|
| Villar-Rodríguez E, Cano-Melle C, Marin-Marin L, Parcet MA, Avila C | 2023 | Structural (VBM), functional (fMRI) and behavioral data relative to the research article "What happens to the inhibitory control functions of the right inferior frontal cortex when this area is dominant for language?" | https://doi.org/10.6084/m9.figshare.22047299.v2 | figshare, 10.6084/m9.figshare.22047299 |

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
