## [Editor Report · eLife assessment]

This study has **important** implications for theoretical proposals concerning how language lateralization affects the lateralization of other cognitive functions. The methods are **solid**, with an appropriate selection of cognitive control tasks that share homotopic regions of the brain with language, comparing participants with typical and atypical organization of language. The participants included in the study were mainly bilinguals, a population previously reported to have a more bilateral organization of cognitive control regions than monolinguals, limiting the generalizability of the results to the general population. Despite this limitation, the results will be of interest to researchers working of brain plasticity and development, in addition to those interested in language and cognitive control.

---

## [Referee Report · Reviewer #1 (Public Review)]

The authors' aim was to test to what extent atypical organization of language is associated with a mirrored brain organization of other cognitive functions. In particular, they focused on the inferior frontal gyri (IFG) by studying the inhibitory control network. This allowed them to directly test support for the Causal hypothesis of hemispheric specialization, arguing for fast sequences of cognitive processes being better performed by a single hemisphere, versus the Statistical hypothesis of lateralization, postulating an independent lateralization of each cognitive function.

Previous studies on this topic did not focus on functions involving homotopic language regions. This limitation is bypassed in this study by assessing inhibition with a Stop-Signal Task which also engages the IFG in the contralateral site to the verb generation task. By studying a combination of structural and functional information, in addition to the activation contrasts, the authors are able to test whether atypical organization is accompanied by stronger interhemispheric connectivity. Although relying mainly on correlations and lacking important methodological information that may be critical to understand the reported effects, the results are quite straightforward. However the bilingual/monolingual status and gender of the participants is not reported which might affect the relationship between language and inhibitory control.

The conclusions of the paper are supported by the data. With their design, the authors observed that, as a group, individuals with atypical organization show a mirror organization of the whole inhibitory network to the contralateral site, supporting the Causal hypothesis at the group level. However, individual data support the Statistical hypothesis, since the segregation between language and inhibition was not observed in all individuals and a variety of configurations in bilateral and bilateral organisation of language and inhibition were also observed.

The results of this study have important implications for our understanding of the independence of different cognitive functions, which is crucial when addressing brain damage and rehabilitation. This aspect also indirectly speaks to researchers interested in evolution and in bilingualism and its relation to cognitive control. These aspects are not discussed but incorporating them would broaden the interest of the paper beyond the current implications mentioned.

---

## [Referee Report · Reviewer #2 (Public Review)]

Language skills are traditionally associated with a network of brain regions in the left hemisphere. In this intriguing study, Esteban Villar-Rodríguez and collaborators examined whether atypical hemispheric lateralization for language determines the functional and structural organisation of the network for inhibitory control as well as its relationship with schizotypy and autistic spectrum traits. The results suggest that individuals who have atypical lateralisation of the language function have also an atypical (mirrored) lateralisation of the inhibitory control network, compared to the typical group (individuals with left-lateralised language function). Furthermore, the atypical organization of language production is associated with a greater white matter volume of the corpus callosum, and atypical lateralization of inhibitory control is related to a higher interhemispheric functional coupling of the IFC, suggesting a link between atypical functional lateralisation (language and inhibitory control) and structural and functional changes in the brain.

This study also provides interesting evidence on how atypical language lateralisation impacts some aspects of language behaviour (reading), i.e., atypical lateralization predicts worse reading accuracy. Furthermore, the results suggest an association between atypical lateralization and increased schizotypy and autistic traits.

---

## [Author Response]

The following is the authors’ response to the original reviews.

**Reviewer #2 (Recommendations For The Authors):**
The evidence provided in this study reflects important discoveries on language lateralisation and most of the conclusions of this paper are supported by evidence. However, there are several areas regarding the characteristics of participants tested, hypotheses/predictions and the type of analysis, that need to be clarified and/or corrected.1. There is a substantial disconnection between the introduction and the methods/results section.One reason is because of lack of consistency. One example refers to the fact that, in the introduction, only IFC is mentioned. However, the analyses carried out to examine neural activity in different groups focused on IFC as well as other brain regions related to inhibitory control. However, these areas were not mentioned at all in the introduction.Second and related to the above, the rationale for conducting certain types of analyses is not specified. Some brain analyses focus on IFC only. Instead, other analyses focus on several areas.Another weakness is that there is not sufficient detail regarding the hypotheses/predictions and the specific types of analyses chosen to test these hypotheses/predictions. For example, there is no mention of resting state fMRI data in the introduction, but later we discover that this type of data was collected and analyzed. Even a brief mention of the inclusion of resting state data in the introduction would be beneficial. Along the same lines, by reading the methods section we find out that VBM analyses were conducted. But it is unclear why. What was the purpose of this data analysis? This should be clarified briefly in the introduction and then in the methods section. It remains unclear why resting state results would be particularly informative for addressing the research question of this study. Task-related brain connectivity seems a more appropriate choice. Additionally, it is not explained what comparisons and outcomes would be informative/expected to distinguish between the two mentioned competing hypotheses. This should be made clear.Another aspect that lacks clarity is the authors' predictions when investigating the relationship "between the lateralization of both functions and inter-hemispheric structural-functional connectivity, as well as with behavioural markers of certain clinical conditions that have been related with atypical lateralization". The hypotheses are completely omitted in this section.

Thank you for bringing this to our attention. We concur with Reviewer #2 that our introduction was somewhat lacking in detail and assumed too much prior knowledge on the part of the reader. This, together with a lack of a clear presentation of our tested hypotheses, made the introduction have a poor connection with both the results and discussion sections, which hindered the understanding of the paper.

As a result, we have made some additions to enhance the exposition of the following areas: (1) the causal and statistical hypotheses of lateralization (Lines 55-65); and (2) the hypotheses regarding subclinical markers of neurological disorders and the corpus callosum (Lines 90-104).

Furthermore, we have extensively revised the final paragraph of the introduction (Lines 105-121) to provide a clearer and more coherent linkage between the drivers presented during the introduction, our hypotheses, and the subsequent analyses.

2. It is important to provide more information on the language background of the participants. Were the participants in this study Catalan-Spanish bilinguals? If so, it is crucial for the authors to mention this.

Language background of the participants has been added to the corresponding section (Lines 138-145).

In fact, previous studies, including several publications from the authors themselves (Garbin et al., 2010; Rodríguez-Pujadas et al., 2013; Anderson et al., 2018), have shown that there are qualitative differences between bilinguals and monolinguals in the neural circuitry underlying executive control. Across all these studies, it was consistently reported that bilingual individuals, when engaged in non-linguistic inhibitory control tasks, recruited a broader network of left-brain regions associated with language control, including the left IFC, in comparison to monolingual individuals. If the participants in this study were indeed bilinguals, it raises concern if the aim of the study is to generalize the conclusions on lateralization effects beyond the bilingual population.Rodríguez-Pujadas, A., Sanjuán, A., Ventura-Campos, N., Román, P., Martin, C., Barceló, F., … & Ávila, C. (2013). Bilinguals use language-control brain areas more than monolinguals to perform non-linguistic switching tasks. PLoS One, 8(9), e73028.Garbin, G., Sanjuan, A., Forn, C., Bustamante, J. C., Rodríguez-Pujadas, A., Belloch, V., ... & Ávila, C. (2010). Bridging language and attention: Brain basis of the impact of bilingualism on cognitive control. NeuroImage, 53(4), 1272-1278.Anderson, J. A., Chung-Fat-Yim, A., Bellana, B., Luk, G., & Bialystok, E. (2018). Language and cognitive control networks in bilinguals and monolinguals. Neuropsychologia, 117, 352-363.

Indeed, we have thoroughly reported that, when compared to monolinguals, bilinguals exhibit a significant implication of left brain regions during switching and inhibition tasks. So, this is a legitimate concern. Unfortunately, the society from which our participants were drawn is primarily bilingual, encompassing both active and passive bilinguals. The monolingual sample in those previous studies consisted of university students originating from predominantly monolingual regions of Spain. Given this context, it is unsurprising that the current study has a rather limited number of monolinguals (n=8), with only 2 displaying atypical language lateralization. Thus, we cannot provide a reliable answer to the role of bilingualism status in our data. Consequently, we have included a comment on this limitation on the discussion (Lines 504-512).

3. Regarding the methods section, I have the following specific queries. The first is about the control condition in the verb generation task. I find it puzzling that the 'task' and 'control' conditions differ in terms of the number of words uttered. Could the authors please provide further clarification on this?

Thank you for raising this question. Regarding the control condition, it is important to note that the design of this task drew inspiration from previously published verb generation tasks for fMRI (Benson et al., 1999; Fitzgerald et al., 1997) and PET (Petersen et al., 1988). In the fMRI tasks, a fixation cross served as the control condition, while the PET study used word repetition as the control. We acknowledged that a mere fixation cross might not adequately control for the movement and visual-related activations inherent in the verb generation task. Conversely, word repetition could potentially engage the default mode network due to the repetition of the same simple task, which might not be suitable for a control condition, and it could be overly linguistic because it involves a word. Consequently, we aimed to strike a balance by employing a control condition that consisted of reading letters. This approach allowed us to control for movement and vision factors without invoking semantics. Thus, after careful consideration, we ultimately opted on the reading of two letters to equate the response to the vocalization length of generating a verb.

Although we understand the concern of single vs. two vocalizations, it is worth emphasizing that this version of the verb generation task had undergone prior testing to assess its suitability for determining language lateralization in both healthy and clinical populations (Sanjuan et al., 2010). In fact, this task has been an integral component of our lab’s standard presurgical assessment protocol, which has been used for nearly two decades in individually evaluating language function in over 500 patients with central nervous system lesions.

Benson, R. R., Fitzgerald, D. B., Lesueur, L. L., Kennedy, D. N., Kwong, K. K., Buchbinder, B. R., Davis, T. L., Weisskoff, R. M., Talavage, T. M., Logan, W. J., Cosgrove, G. R., Belliveau, J. W., & Rosen, B. R. (1999). Language dominance determined by whole brain functional MRI in patients with brain lesions. Neurology, 4(52), 798–809.

Fitzgerald, D. B., Cosgrove, G. R., Ronner, S., Jiang, H., Buchbinder, B. R., Belliveau, J. W., Rosen, B. R., & Benson, R. R. (1997). Location of Language in the Cortex: A Comparison between Functional MR Imaging and Electrocortical Stimulation. AJNR Am J Neuroradiol, 18, 1529–1539.

Petersen, S. E., Fox, P. T., Posner, M. I., Mintun, M., & Raichle, M. E. (1988). Positron emission tomographic studies of the cortical anatomy of single-word processing. Nature, 331(18), 585–589.

Sanjuán, A., Bustamante, J. C., Forn, C., Ventura-Campos, N., Barrós-Loscertales, A., Martínez, J. C., Villanueva, V., & Ávila, C. (2010). Comparison of two fMRI tasks for the evaluation of the expressive language function. Neuroradiology, 52(5), 407–415. https://doi.org/10.1007/s00234-010-0667-8

Second, it is mentioned that some participants were excluded from different tasks due to technical issues or time constraints. It is important to ensure that all the results can be attributed to the exact same sample of participants across all tasks.

We absolutely agree that excluding participants can be problematic when presenting the results of multiple sets of analyses. Therefore, we repeated all analyses while excluding the 7 participants that lacked resting-state data. All results remained virtually identical, with a few minor exceptions:

1. Region-wise analysis of the stop-signal task: Hemisphere × Group effect in the preSMA region is significant (uncorrected P = 0.019), but it does not survive Bonferroni correction (corrected P = 0.076)

2. Voxel-wise analysis of the stop-signal task: The Thalamus + STN and Caudate clusters are significant at the voxel level, but do not survive the cluster-based FWE correction. They do survive FDR correction, though.

3. Correlation between SPQ score and LI of the stop-signal task: This correlation weakens just behind statistical significance, with a P value of 0.053.

4. Correlation between reading variables and LIs of both tasks: Severe drops in P values are evident between both LIs and reading length accuracy (P = .111 and .133), as well as between verb generation LI and reading familiarity accuracy (P = .111). However, the association between the stop-signal LI and the reading length time is now significant (r = −.229, P = .042).

According to this, we have included this statement in the methods section: (Lines 218-220).“It is important to highlight that the exclusion of these seven participants across all analyses does not notably impact the overall results.“

It is unclear how the authors have estimated the RTs results from the practice trials. This requires more explanation. Also, why was the median used for the Go Reaction Time instead of the mean, when calculating the individual SSRT?

We adapted the procedure used by Xue et al. (2008), implementing their approach to calculate SSRT. This has been elaborated further (Lines 227-230), together with the use of practice trials (Lines 233-236).

Xue, G., Aron, A.R., and Poldrack, R.A. (2008). Common Neural Substrates for Inhibition of Spoken and Manual Responses. Cerebral Cortex 18, 1923–1932. 10.1093/CERCOR/BHM220.

On a final note, information about the different types of pre-processing and data analysis is all reported in the same paragraph. I think using subsections would increase the intelligibility of the section.

Thank you for this suggestion. We have added subsections in both the ‘image processing’ and ‘statistical analyses’ sections.

4. Data analysis and Interpretation of the results. It is unclear how the mean BOLD signal was extracted to conduct ROI analysis (Marsbar?).

Thank you for ponting this out. Indeed, we were not very accurate in the description of this procedure. We extracted the first eigenvariate via the VOI function within SPM12. This has been included in Lines 291-293.

I feel uneasy about the way results are corrected for multiple comparisons. For instance, it is mentioned that in the ROI analysis, all p-values were FDR-corrected for four comparisons, but it is unclear why. The correct procedure for supporting conclusions about the effect of specific brain would be to have 'brain region' (n=4) as another within-subject factor. Furthermore, the one-tailed correlation is appropriate but only when testing for the possibility of a relationship in one direction and completely disregarding the possibility of a relationship in the other direction. However, this does not seem to be the case here (see Introduction), so a two-tailed correlation would be more appropriate.

We agree with Reviewer #2 that presenting this analysis as a single MANOVA that includes a ‘Region’ factor is a more accurate approach. Consequently, we have made the aforementioned correction in the methods section (Lines 357-364) and the results section (Lines 395-406). The LI-LI one-tailed correlation was also changed to a two-tailed correlation in the methods section (Line 383), the results section (Line 417), and Figure 2 (Line 886).

I am quite confused about using the term interhemispheric connectivity to refer to the volume of the genu, body and splenium of the corpus callosum. In fact, the volumes of genu, body and splenium of the corpus callosum do not reflect a measure of how strongly RH and LH IFC are connected to each other.

We agree that using the term ‘interhemispheric connectivity’ when referring to callosal volume may be somewhat misleading. We have replaced every instance of this terminology throughout the paper.

Furthermore, it is unclear why in a set of analyses (ROI and whole brain analyses) the authors focus on brain responses in different ROIs but instead, in connectivity measures the focus is only on IFC.

Our initial rationale was to focus on regions that are prominently involved in language, particularly the IFC, for examining inter-hemispheric connectivity at rest.

However, upon more careful consideration, it is true that the preSMA is also implicated in the language network (Labache et al., 2018), and certain studies have reported an impact of STN stimulation on specific language skills (for a review, see Vos et al., 2021). Consequently, we have incorporated these two regions into the resting-state analysis, along with subsequent correlations with LIs (Table 1 and Lines 118, 321-322 & 449-452).

Labache, L., Joliot, M., Saracco, J., Jobard, G., Hesling, I., Zago, L., Mellet, E., Petit, L., Crivello, F., Mazoyer, B., & Tzourio-Mazoyer, N. (2018). A SENtence Supramodal Areas AtlaS (SENSAAS) based on multiple task-induced activation mapping and graph analysis of intrinsic connectivity in 144 healthy right-handers. Brain Structure and Function 2018 224:2, 224(2), 859–882. https://doi.org/10.1007/S00429-018-1810-2

Vos, S. H., Kessels, R. P. C., Vinke, R. S., Esselink, R. A. J., & Piai, V. (2021). The Effect of Deep Brain Stimulation of the Subthalamic Nucleus on Language Function in Parkinson’s Disease: A Systematic Review. Journal of Speech, Language, and Hearing Research, 64(7), 2794–2810. https://doi.org/10.1044/2021_JSLHR-20-00515

Minor corrections/comments:It is unclear why in figure caption 1, the conjunction maps are mentioned even if formal conjunction analysis was not conducted.

This poor choosing of words has been replaced to ‘overlapping maps’.

Line 382. VHMC should be VMHC.

Fixed. Thank you.

Line 334. This sentence and especially its relationship with the results is not clear at all. What do you mean by 'This finding is consistent with previous reports showing that cognitive deficits appear only in specific cognitive domains'?

This has been clarified (Lines 521-525).